# Effects of Developmental Failure of Swallowing Threshold on Obesity and Eating Behaviors in Children Aged 5–15 Years

**DOI:** 10.3390/nu14132614

**Published:** 2022-06-24

**Authors:** Yuko Fujita

**Affiliations:** Division of Developmental Stomatognathic Function Science, Department of Health Promotion, Kyushu Dental University, Kitakyushu 803-8580, Japan; y-fujita@kyu-dent.ac.jp

**Keywords:** eating behaviors, obesity, swallowing threshold, number of chewing cycle, children

## Abstract

Background: The aim of the present study was to identify factors related to developmental failure of swallowing threshold in children aged 5–15 years. Methods: A total of 83 children aged 5–15 years were included in this study. A self-administered lifestyle questionnaire was completed, along with hand grip strength and oral function tests. Swallowing threshold was determined based on the concentration of dissolved glucose obtained from gummy jellies when the participants signaled that they wanted to swallow the chewed gummy jellies. Developmental failure of swallowing threshold was defined as glucose concentrations in the lowest 20th percentile. After univariate analysis, multivariate binary logistic regression analysis was used to identify factors associated with developmental failure of swallowing threshold. Results: Hand grip strength was significantly correlated with masticatory performance (*r* = 0.611, *p* < 0.01). Logistic regression analysis revealed factors related to developmental failure of swallowing threshold, i.e., overweight/obesity (Odds ratio) (OR) = 5.343, *p* = 0.031, 95% CI = 1.168–24.437) and eating between meals at least once a day (OR = 4.934, *p* = 0.049, 95% CI = 1.004–24.244). Conclusions: Developmental failure of swallowing threshold was closely associated with childhood obesity in 5- to 15-year-old children.

## 1. Introduction

Eating habits acquired in early childhood can change over time based on personal experience and learning [1,2]. Eating habits acquired in childhood or adolescence may persist into adulthood [3]. Poor dietary habits established during childhood that persist into adulthood increase the risk of obesity and related adverse consequences [4,5,6].

In previous studies, early childhood obesity was associated with a higher risk of adult metabolic syndrome [7,8]. Eating behaviors such as frequent consumption of fast food and sugar-sweetened beverages are associated with the development of childhood obesity [9]. In addition, factors such as eating frequency, amount, and occasions may interact to cause obesity [5,7]. Other studies indicated that early modification of poor eating habits could improve health and decrease the future risk of metabolic syndrome, type 2 diabetes, and atherosclerotic cardiovascular diseases [10,11].

A relationship between obesity and eating speed was reported recently. Eating speed was associated with the incidence of metabolic syndrome in adults [12,13], while eating quickly was associated with increased body mass index (BMI) in young adults [14]. Similar results have also been reported in children [15,16,17]. In addition, chewing well before swallowing might be an effective method for reducing food intake and facilitating healthy weight management in preschool children [18] and adults [19].

These findings suggest that appropriate eating behaviors should be acquired within the critical early time period. However, the number of chewing cycles, chewing time, and chewing rate have not been analyzed in the context of the swallowing threshold, and it remains unclear when these functions are acquired. We hypothesized that these functions would be acquired relatively early in childhood. 

Additionally, based on previous studies, we hypothesized that developmental failure of swallowing threshold in children would be associated with eating fast, chewing frequently, and other habits [17,20]. This developmental failure could involve physiological problems including obesity. Identifying the factors associated with developmental failure of swallowing threshold may aid the development of strategies for improving it.

The primary purpose of this study was to clarify differences in swallowing thresholds and other oral functions according to age, as well as when functions related to the swallowing threshold are acquired. A secondary aim was to determine factors related to the developmental failure of swallowing threshold in children aged 5–15 years. In this study, swallowing threshold was determined based on the concentration of dissolved glucose obtained from gummy jellies when the participants signaled that they wanted to swallow the chewed gummy jellies.

## 2. Materials and Methods

### 2.1. Participants

This study was conducted in accordance with the Declaration of Helsinki and was approved by the Human Investigation Committee of Kyushu Dental University (approval number: 18–37). All participants and their parents/guardians provided written informed consent for participation in the study. The participants were recruited following an initial examination at two private dental clinics in Japan. The inclusion criteria were as follows: aged 5–15 years, normal language comprehension, cooperative behavior, and no current dental diseases or complications. The exclusion criteria included systemic disturbances causing swallowing impairment, obvious facial asymmetry that could affect the measurements, soft tissue abnormalities, temporomandibular joint dysfunction, and dental or structural irregularities.

Using software (PS: Power and Sample Size Calculation, available from the Vanderbilt University’s website), sample size was calculated. The study was based on a continuous response variable ranging from normal swallowing threshold to developmental failure of swallowing threshold; four normal participants were recruited for every participant with the developmental failure of swallowing threshold. In a previous study, the data for each group were normally distributed, with a standard deviation (SD) of swallowing threshold of 40 mg/dL [21]. Based on a true difference of 40 mg/dL between the developmental failure and normal groups, 14 developmental failure and 56 normal participants were required to reject the null hypothesis (i.e., the population means of the developmental failure and normal groups are equal) with a power of 0.9. The type I error probability for the null hypothesis was 0.05.

The participants were divided into 11 groups based on their chronological age, and each group was further divided into two subgroups based on sex. Details of the participants are shown in Table 1. 

### 2.2. Questionnaire

The survey solicited the following information: demographic characteristics (sex and age), eating habits, physical activity, and sleep [21].

### 2.3. Anthropometry and Dental Examination

Height and body weight were measured in the consultation room of the clinic. Height was measured to an accuracy of ±0.1 cm using a portable digital stadiometer (AD-653; A&D, Tokyo, Japan), with the head in the Frankfort plane, whereas body weight was measured with an accuracy of 0.1 kg [22]. Rohrer and Kaup indices were calculated from the height and weight. 

During intraoral examination, the number of erupted teeth and the decayed, missing, and filled teeth (DMFT) index were recorded for each patient [23].

### 2.4. Hand Grip Strength

Hand grip strength is generally used as an index of muscle weakness in the diagnosis of sarcopenia [24]. Therefore, hand grip strength measurement was performed for evaluating systemic muscle strength in the participants. A portable grip strength meter (T-2288; Toei Light Co., Ltd., Saitama, Japan) was used to measure hand grip strength. Participants were asked to stand and hold the dynamometer in their hand with the arm parallel to the body, but without squeezing the arm against the body. Hand grip strength (kg) was measured twice for each hand (alternately) with a 30 s interval between trials. The highest value from either the left or right hand was recorded as the grip strength [20].

### 2.5. Maximum Occlusal Force

Maximum occlusal force was measured using a portable occlusal force meter (GM10; Nagano Keiki Co., Ltd., Tokyo, Japan). The participants were instructed to bite down with maximal voluntary muscular effort using their first molars. Maximum occlusal force was measured on each side, with a 30 s interval between bite measurements. The larger of the values from the left and right sides was recorded as the maximum bite force [21].

### 2.6. Masticatory Performance

Masticatory performance was determined by measuring the concentration of dissolved glucose obtained from a cylindrical gummy jelly (GLUCOLUMN; GC Co., Ltd., Tokyo, Japan) using a glucose-measuring device (GLUCO SENSOR GS-II; GC Co., Ltd., Tokyo, Japan). Prior to the test, the participants were instructed regarding the chewing movements and mouth rinsing procedures to prevent swallowing. The participants were then instructed to chew the gummy jelly for 20 s freely. After chewing, the participants were asked to take 10 mL of distilled water into their mouth and spit out the gummy jelly and distilled water into a filter cup. The glucose concentration in the filtrate (mg/dL) was measured using a dedicated device [20].

### 2.7. Swallowing Threshold

Following evaluation of the masticatory performance, an assessment of the swallowing threshold was performed using the test gummy jellies (GLUCOLUMN; GC Co. Ltd.). Variables related to swallowing (i.e., the number of chewing cycles, chewing time, chewing rate, and glucose concentration in the filtrate (mg/dL)) were assessed for all participants. Each participant was instructed to chew a gummy jelly freely until feeling the desire to swallow, at which time they were instructed to stop chewing and signal to the examiner that they were ready to expel the gummy jelly. The examiner visually quantified the number of chewing cycles. The time from the onset of chewing to the moment when the participants raised their hands was recorded using a stopwatch [20]. The subsequent steps were the same as those used for the evaluation of masticatory performance. We determined that the glucose concentration in the filtrate was an indicator of the swallowing threshold. Participants with low swallowing threshold tend to signal that they want to swallow the gummy jelly quickly, resulting in low glucose concentrations. A glucose concentration in the lowest 20th percentile was defined as developmental failure of swallowing threshold, based on previous studies reporting the criteria for sarcopenia [21,25]. 

### 2.8. Reliability of Measurements

All measurements were performed in duplicate, separated by 30 s rest periods, and the mean values were calculated for the subsequent analyses. All examinations were performed by the same examiner. Data generated during sample collection were assessed for reliability. Random error was evaluated based on intraobserver reliability, which was quantified using the intraclass correlation coefficient (ICC; where 0.8 ≤ ICC ≤ 1.0 corresponded to high reliability) [26].

### 2.9. Statistical Analysis

The Shapiro–Wilk test was used to determine data normality. All continuous data were expressed as means ± SD. The means were compared between two groups using a two-tailed *t*-test or the Mann–Whitney *U* test. The Kruskal–Wallis test was used for comparisons between more than two groups. Pearson’s correlation coefficient was used to determine associations among glucose concentration before swallowing and other variables. The chi-squared test or Fisher’s exact test was used, as appropriate, to compare categorical variables between the normal swallowing threshold and developmental failure of swallowing threshold (lowest 20%) groups. Binary logistic regression analysis with the forward selection (conditional) method was used to identify factors predicting developmental failure of swallowing threshold. Independent variables that were significant in the univariate analyses were included. Categorical variables were coded appropriately before being entered into the model. Adjusted odds ratios (ORs) and 95% confidence intervals (CIs) were calculated for the low swallowing threshold groups. A *p*-value < 0.05 was considered statistically significant. All data were analyzed using SPSS Statistics for Windows software (version 23.0; IBM Corp., Armonk, NY, USA).

## 3. Results

The ICCs for all measurement items ranged from 0.84 to 0.91, indicating a high degree of intraobserver reliability. 

The anthropometric parameters and dental examination results are shown in Table 1 according to age and sex. The 15-year-old males were significantly taller than females of the same age (*p* < 0.05). The DMFT index in 11-year-old females was significantly higher than in males of the same age (*p* < 0.05).

Hand grip strength and oral functional parameters are shown in Table 2 according to age and sex. The hand grip strengths in 5-, 10-, 12-, 14-, and 15-year-old males were significantly greater compared to females of the same age (all *p* < 0.05). 

When considering the results of comparing the mean scores of males and females for all measurement items by age, hand grip strength in 14–15-year-old participants was significantly greater compared to 5–7-year-old participants (all *p* < 0.05). Hand grip strength increased with age. The maximum occlusal force was greatest in 15-year-old participants and was significantly greater than in those aged 5–7 years (all *p* < 0.05). Masticatory performance was lowest in 5-year-old participants and was significantly lower than in those aged 12, 14, and 15 years (all *p* < 0.05). The number of chewing cycles was highest in 9-year-old participants and was significantly higher than in those aged 6 and 14 years (both *p* < 0.05). The change in chewing time with age was similar to the change in the number of chewing cycles. The swallowing threshold was highest in 15-year-old participants, followed by the 8-year-old participants; both were significantly higher than in those aged 6 years (*p* < 0.05).

The results of Pearson’s bivariate correlation analyses are shown in Table 3. Hand grip strength was significantly and positively correlated with masticatory performance (*r* = 0.611, *p* < 0.01) and was strongly correlated with maximum occlusal force (*r* = 0.791, *p* < 0.01). Swallowing threshold was strongly correlated with masticatory performance (*r* = 0.727, *p* < 0.01) and was significantly and positively correlated with the number of erupted teeth, maximum occlusal force, and number of chewing cycles (*r* = 0.432, *p* < 0.01; *r* = 0.493, *p* < 0.01; *r* = 0.465, *p* < 0.01, respectively). The number of chewing cycles was strongly correlated with chewing time (*r* = 0.801, *p* < 0.01). The correlation between age and swallowing threshold was weak but significant (*r* = 0.362, *p* < 0.01; Figure 1).

Table 4 summarizes the data on developmental failure of swallowing threshold based on demographic and health-related variables. Participants with developmental failure of swallowing threshold were more likely to be overweight or obese, eat between meals at least once per day, and have low or no physical activity (*p* = 0.014, *p* = 0.021, and *p* = 0.006, respectively).

Table 5 summarizes the data on developmental failure of swallowing threshold based on age, height, body weight, dental status, hand grip strength, and masticatory function. Swallowing threshold, age, number of erupted teeth, maximum occlusal force, masticatory performance, number of chewing cycles, and chewing time in the develop-mental failure of swallowing threshold group were significantly lower compared to the normal swallowing threshold group (all *p* < 0.05). The mean DMFT index in the developmental failure swallowing threshold group was significantly higher compared to the normal swallowing threshold group (*p* < 0.05). The mean chewing time to swallow in the normal swallowing threshold group was 20.91 s. Masticatory performance was similar between the groups.

Table 6 shows the predictors of developmental failure of swallowing threshold, as revealed by logistic regression. Overweight or obese individuals had higher odds of developmental failure of swallowing threshold (OR = 5.343, *p* = 0.031, 95% CI = 1.168–24.437). Eating between meals once or more a day was also associated with higher odds of developmental failure of swallowing threshold (OR = 4.934, *p* = 0.049, 95% CI = 1.004–24.244).

## 4. Discussion

In this study, patterns of swallowing threshold in children according to age were clearly different from those of handgrip strength, maximum occlusal force, and masticatory performance. Maximum occlusal force and masticatory performance tended to increase with age, but the rates of increase between the ages of 12 and 15 years were not as high as those for hand grip strength. These results suggest that maximum occlusal force and masticatory performance do not grow rapidly during this period. 

When considering the results of univariate analysis, hand grip strength was significantly and positively correlated with masticatory performance as well as maximum occlusal force. The European Working Group on Sarcopenia in Older People 2 (EWGSOP2) reported that low activity and malnutrition cause secondary sarcopenia not only in the elderly but also in children [24]. Sarcopenia may be associated with developmental failure of masticatory function in childhood. 

We found that age had a weak correlation with the swallowing threshold. At the age of 8 years, multiple participants had swallowing thresholds that reached or surpassed those of 15-year-old individuals (Figure 1). A recent study of individuals aged 20–79 years reported that the swallowing threshold values peaked at 40 s, which was close to the peak time for 15-year-old females in this study [21]. Therefore, the swallowing threshold is unlikely to be age-dependent, and we derived a “unified” threshold of developmental failure, i.e., a threshold applicable to all ages. The number of chewing cycles and chewing time tended to gradually decrease after peaking at the age of 9 and 8 years, respectively.

In the logistic regression analysis, obesity was significantly associated with developmental failure of swallowing threshold. A previous study reported that children who did not chew well had a significantly higher likelihood of overweight compared to the reference group of children aged 5–6 years. To assess the degree of chewing, the parents in that study answered the following self-administered questionnaire item: “How well does your child chew foods while eating?” [18]. Another study objectively evaluating the swallowing threshold reported that a lower threshold was associated with a higher BMI among preschool children aged 3–5 years. They suggested that a higher BMI was associated with a lower number of chewing cycles and a shorter chewing time [17]. A study of young adults (in their twenties) also reported that BMI was negatively correlated with the total number of chews and duration of chewing but did not correlate significantly with the chewing rate [14]. Consistent with these findings, our results showed that the number of chewing cycles and chewing time in participants with developmental failure of swallowing threshold were significantly lower compared to those with a normal swallowing threshold. However, chewing rate was not significantly different between the two groups. These findings suggested that overweight/obesity is more prevalent among children who chew their food less, and over a shorter period of time, prior to swallowing, regardless of the chewing rate.

Several studies have reported that the causes of obesity are multi-faceted and include interactions among eating behaviors, obesogenic environments, genetics, and physical inactivity [5,7]. This may be indirectly associated with our finding that a significantly higher percentage of participants with developmental failure of swallowing threshold reported a lack of physical activity according to univariate analysis, although a causal relationship could not be determined.

In the logistic regression analysis, eating between meals at least once per day was also significantly associated with developmental failure of swallowing threshold. Chewing food activates the hypothalamic histamine nervous system and suppresses appetite through the satiety center in animals [27,28,29]. Children with developmental failure of swallowing threshold are less likely to feel sated, so it is more likely that they will eat between meals at least once per day. Several studies reported that consumption of low-quality snacks was associated with increased prevalence of obesity among children [30,31,32]. Our results indicate that this may be explained in terms of developmental failure of swallowing threshold. A study of Japanese elementary school children reported that their snacking habits were influenced by paternal eating habits [33]. Another study suggested that parents could play an important role in the control of school-age children’s food intake and choices [6]. National surveys of food intake in the United States reported that daily energy intake in children aged 2 to 18 increased significantly from 1839 kcal/day to 2023 kcal/day between 1977–1978 and 2003–2006 [34]. In addition, another study reported that children aged 2 to 11 consume extra energy and sugars in their diets but insufficient Vitamin D, calcium, and potassium [35]. This may be applicable to all eating behaviors, i.e., not just snacking.

We further considered strategies to improve developmental failure of swallowing threshold. First, the chewing rate until swallowing was not significantly different between participants with developmental failure and normal swallowing thresholds. Therefore, it may not be necessary to control the chewing rate. Second, prolonging the chewing time until swallowing may also not be helpful for preventing developmental failure of swallowing threshold; even after the participants with developmental failure of swallowing threshold chewed the gummy jelly for 20 s, masticatory performance was significantly lower (97.94 ± 24.97 mg/dL) than that of participants with normal swallowing thresholds (134.19 ± 40.75 mg/dL). The lower masticatory performance may be associated with the younger age, weaker occlusal forces, a higher DMFT index, and fewer erupted teeth in participants with developmental failure of swallowing threshold. If these problems are resolved, developmental failure of swallowing threshold may improve. However, as it is difficult to solve these problems immediately, increasing the number of chewing cycles may be the key to improving developmental failure of swallowing threshold. Considering that it is a critical period of development, the appropriate number of chewing cycles and chewing time until swallowing should be established by the age of 9 years.

Our study had some limitations. First, The National Health Examination Survey among 12- to 17-year-old US adolescents reported that BMI in the adolescents with a concave facial profile was higher than that in the adolescents with a straight facial profile [36]. Another study reported that a higher BMI was associated with a lower bite force in children aged 8–10 years [37]. These results suggest that facial profile type in children, especially Class III malocclusion, could be a factor associated with developmental failure of swallowing threshold. However, in this study, relationships between the type of malocclusion and the characteristics of the swallowing threshold were not able to be clarified, because participants’ malocclusions were not evaluated in detail. In the future, it is necessary to clarify the relationship between malocclusion and obesity and developmental failure of swallowing threshold in children. Second, the number of participants was small for performing logistic regression analysis. Among the patients who visited two dental clinics, the number of patients who met all of the conditions we set was very limited. Therefore, this study was performed with the minimum required number of participants calculated by the power analysis. It may be that a highly accurate model will be built in studies with higher numbers of participants. Third, it used a cross-sectional design, which precluded determination of causal relationships between developmental failure of swallowing threshold and the variables included in the logistic regression analyses. Longitudinal studies are needed to investigate the relative influence of factors associated with developmental failure of swallowing threshold. Additionally, the diet- and lifestyle-related questionnaires evaluated only children’s problems; future studies may benefit from evaluating the parents’ eating habits and dietary education.

## 5. Conclusions

We found that the relationship between swallowing threshold and age was unclear, whereas the number of chewing cycles and chewing time tended to gradually decrease after reaching a peak at the age of 9 and 8 years, respectively. An appropriate number of chewing cycles and chewing time until swallowing should be established by 9 years of age. Developmental failure of swallowing threshold was closely associated with childhood obesity and eating between meals at least once per day among the 5- to 15-year-old individuals.

## Figures and Tables

**Figure 1 nutrients-14-02614-f001:**
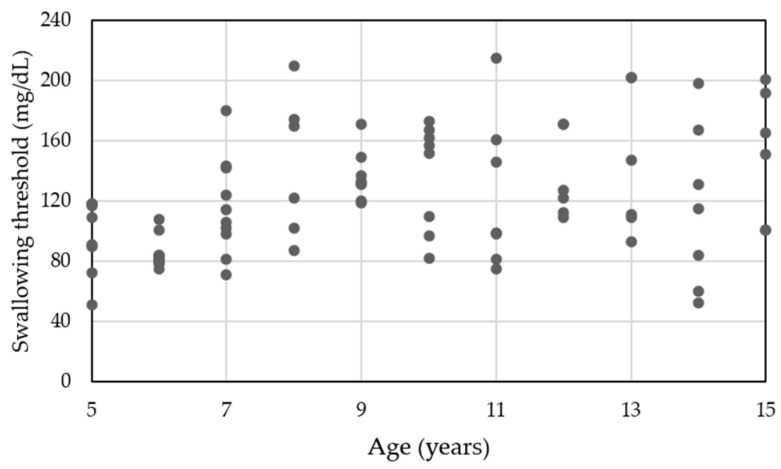
Correlation between age and swallowing threshold.

**Table 1 nutrients-14-02614-t001:** Anthropometric parameters and dental examination results by age and sex.

Age (Years)	Sex (*n* = 83)	Height (m)	Body Weight (kg)	Kaup Index/Rohrer Index	Number of Erupted Teeth	DMFT Index
5	M (*n* = 4)	1.18 ± 0.08	17.75 ± 0.96	13.01 ± 2.09	20.25 ± 0.50	2.50 ± 2.38
	F (*n* = 4)	1.17 ± 0.14	16.75 ± 1.26	12.86 ± 2.81	20.00 ± 0.00	7.75 ± 7.50
6	M (*n* = 5)	1.13 ± 0.03	20.40 ± 1.67	143.81 ± 19.83	19.40 ± 1.34	7.00 ± 5.39
	F (*n* = 6)	1.13 ± 0.04	20.67 ± 1.37	144.30 ± 12.77	20.00 ± 0.00	8.00 ± 5.02
7	M (*n* = 4)	1.29 ± 0.09	26.25 ± 4.99	122.24 ± 8.08	21.25 ± 1.50	5.50 ± 2.52
	F (*n* = 6)	1.22 ± 0.07	22.50 ± 2.07	126.02 ± 21.46	21.50 ± 1.76	6.00 ± 3.16
8	M (*n* = 3)	1.21 ± 0.06	21.33 ± 3.06	120.54 ± 16.38	23.67 ± 0.58	2.67 ± 2.08
	F (*n* = 3)	1.27 ± 0.07	24.67 ± 4.16	119.10 ± 13.05	23.67 ± 0.58	5.00 ± 4.36
9	M (*n* = 4)	1.39 ± 0.05	32.88 ± 8.53	122.06 ± 20.64	24.00 ± 0.00	6.00 ± 0.82
	F (*n* = 4)	1.32 ± 0.07	29.75 ± 6.90	127.73 ± 18.58	23.25 ± 1.50	3.25 ± 3.20
10	M (*n* = 3)	1.38 ± 0.03	29.50 ± 0.50	111.56 ± 5.50	23.67 ± 0.58	6.67 ± 1.53
	F (*n* = 5)	1.32 ± 0.07	26.60 ± 4.55	115.37 ± 5.97	23.20 ± 0.84	7.60 ± 3.91
11	M (*n* = 4)	1.36 ± 0.09	29.25 ± 2.99	121.96 ± 16.85	24.00 ± 2.94	2.00 ± 2.71
	F (*n* = 3)	1.41 ± 0.01	30.33 ± 1.15	108.18 ± 2.45	23.00 ± 1.00	10.33 ± 1.53 *
12	M (*n* = 3)	1.43 ± 0.04	50.00 ± 7.21	173.45 ± 34.21	25.00 ± 2.65	3.33 ± 4.04
	F (*n* = 3)	1.37 ± 0.10	32.67 ± 8.39	123.82 ± 7.72	25.00 ± 1.00	1.33 ± 1.53
13	M (*n* = 3)	1.64 ± 0.14	49.67 ± 12.70	110.73 ± 4.59	26.67 ± 2.31	4.33 ± 2.08
	F (*n* = 3)	1.54 ± 0.02	45.67 ± 0.58	128.40 ± 4.49	28.00 ± 0.00	6.00 ± 4.36
14	M (*n* = 3)	1.76 ± 0.02	72.33 ± 2.08	131.91 ± 1.75	28.00 ± 0.00	3.67 ± 3.51
	F (*n* = 4)	1.55 ± 0.08	45.25 ± 1.71 *	123.35 ± 17.04	27.00 ± 2.00	1.25 ± 1.50
15	M (*n* = 3)	1.74 ± 0.05	72.00 ± 14.80	135.16 ± 20.15	28.00 ± 0.00	4.67 ± 2.31
	F (*n* = 3)	1.51 ± 0.08 *	47.33 ± 6.81	125.64 ± 13.49	28.00 ± 0.00	5.67 ± 9.81

Data are expressed as mean ± standard deviation. M, male; F, Female; DMFT, decayed, missing, and filled teeth; Differences between males and females within each group were assessed by the Mann–Whitney *U* test. * *p* < 0.05 vs. group-matched males.

**Table 2 nutrients-14-02614-t002:** Hand grip strength and oral functional parameters by age group and sex.

Age (Years)	Sex (*n* = 83)	Hand Grip Strength (kg)	Maximum Occlusal Force (kN)	Masticatory Performance (mg/dL)	Number of Chewing Cycles (N)	Chewing Time (s)	Chewing Rate (s/N)	Swallowing Threshold (mg/dL)
5	M (*n* = 4)	7.50 ± 1.29	0.21 ± 0.04	86.25 ± 12.89	18.25 ± 10.72	15.75 ± 8.26	0.89 ± 0.13	83.00 ± 28.48
	F (*n* = 4)	5.75 ± 0.50 *	0.24 ± 0.08	87.50 ± 3.79	31.00 ± 1.41	26.25 ± 2.22 *	0.85 ± 0.07	108.50 ± 12.97
6	M (*n* = 5)	8.90 ± 1.43	0.24 ± 0.05	99.00 ± 22.03	16.80 ± 1.79	15.60 ± 2.19	0.93 ± 0.14	85.20 ± 13.14
	F (*n* = 6)	9.00 ± 1.26	0.22 ± 0.04	100.83 ± 15.72	16.17 ± 1.47	16.67 ± 3.61	1.02 ± 0.16	84.83 ± 8.11
7	M (*n* = 4)	10.50 ± 1.00	0.27 ± 0.08	104.75 ± 17.73	23.50 ± 7.55	20.00 ± 6.68	0.85 ± 0.08	100.75 ± 14.08
	F (*n* = 6)	9.30 ± 2.92	0.27 ± 0.08	100.33 ± 30.79	25.33 ± 4.41	24.67 ± 5.43	0.97 ± 0.11	126.33 ± 38.14
8	M (*n* = 3)	11.00 ± 4.77	0.36 ± 0.10	101.67 ± 48.40	31.33 ± 8.96	28.33 ± 3.06	0.98 ± 0.43	127.67 ± 43.78
	F (*n* = 3)	11.17± 2.25	0.30 ± 0.01	153.00 ± 49.24	25.00 ± 6.56	22.00 ± 3.61	0.90 ± 0.10	160.67 ± 54.60
9	M (*n* = 4)	15.25 ± 6.54	0.37 ± 0.05	114.50 ± 10.34	36.00 ± 8.60	25.50 ± 3.11	0.73 ± 0.10	129.00 ± 6.06
	F (*n* = 4)	11.00 ± 2.71	0.34 ± 0.06	146.50 ± 40.62	25.00 ± 2.58	21.25 ± 5.56	0.84 ± 0.15	144.00 ± 21.82
10	M (*n* = 3)	24.33 ± 0.58	0.37 ± 0.06	171.33 ± 12.66	22.67 ± 1.53	19.00 ± 1.00	0.84 ± 0.04	162.00 ± 5.00
	F (*n* = 5)	10.30 ± 1.57 *	0.33 ± 0.05	122.60 ± 34.78	26.60 ± 7.89	22.20 ± 7.01	0.83 ± 0.05	122.80 ± 38.30
11	M (*n* = 4)	18.63 ± 6.94	0.36 ± 0.09	159.00 ± 59.35	25.75 ± 4.86	21.50 ± 7.14	0.87 ± 0.34	155.00 ± 48.19
	F (*n* = 3)	11.67 ± 0.58	0.32 ± 0.05	106.33 ± 17.47	19.33 ± 0.58	17.00 ± 1.00	0.88 ± 0.03	85.00 ± 12.49
12	M (*n* = 3)	29.50 ± 5.63	0.46 ± 0.05	176.67 ± 7.57	16.67 ± 3.21	14.33 ± 2.89	0.86 ± 0.03	150.33 ± 35.80
	F (*n* = 3)	16.00 ± 4.36 *	0.34 ± 0.04 *	130.67 ± 9.07 *	20.33 ± 4.93	17.33 ± 4.93	0.85 ± 0.03	120.33 ± 7.64
13	M (*n* = 3)	30.17 ± 11.51	0.49 ± 0.06	136.33 ± 47.72	22.00 ± 5.00	17.00 ± 1.73	0.79 ± 0.13	147.33 ± 54.50
	F (*n* = 3)	21.50 ± 2.18	0.35 ± 0.08	139.67 ± 65.25	27.00 ± 1.73	20.00 ± 1.73	0.74 ± 0.02	140.67 ± 53.13
14	M (*n* = 3)	38.00 ± 3.00	0.44 ± 0.01	143.67 ± 21.96	12.00 ± 5.57	11.24 ± 5.28	0.93 ± 0.02	65.33 ± 16.65
	F (*n* = 4)	24.75 ± 2.75 *	0.38 ± 0.10	167.50 ± 31.42	19.50 ± 2.38	18.75 ± 2.50	0.97 ± 0.17	152.75 ± 37.19 *
15	M (*n* = 3)	43.67 ± 6.43	0.52 ± 0.06	169.33 ± 61.04	18.67 ± 9.02	12.00 ± 4.00	0.71 ± 0.25	117.67 ± 28.87
	F (*n* = 3)	21.67 ± 1.53 *	0.47 ± 0.03	164.33 ± 21.01	25.00 ± 5.20	21.33 ± 3.79	0.86 ± 0.04	186.00 ± 18.73 *

Data are expressed as mean ± standard deviation. M, male; F, female. Differences between males and females within each group were assessed by the Mann–Whitney *U* test. * *p* < 0.05 vs. group-matched male.

**Table 3 nutrients-14-02614-t003:** Bivariate correlation coefficients for the anthropometry, dental status, and masticatory function.

	Age	Height	Body Weight	Number of Erupted Teeth	DMFT Index	Hand Grip Strength	Maximum Occlusal Force	Masticatory Performance	Number of Chewing Cycles	Chewing Time	Chewing Rate
Age	1										
Height	0.871 **	1									
Body weight	0.833 **	0.916 **	1								
Number of erupted teeth	0.885 **	0.815 **	0.778 **	1							
DMFT index	−0.195	−0.207	−0.148	−0.232 *	1						
Hand grip strength	0.795 **	0.848 **	0.903 **	0.762 **	−0.216 *	1					
Maximum occlusal force	0.760 **	0.738 **	0.722 **	0.734 **	−0.339 **	0.791 **	1				
Masticatory performance	0.580 **	0.471 **	0.498 **	0.587 **	−0.375 **	0.611 **	0.659 **	1			
Number of chewing cycles	−0.104	−0.101	−0.236 *	−0.026	−0.064	−0.203	0.049	−0.028	1		
Chewing time	−0.238 *	−0.295 **	−0.396 **	−0.156	−0.052	−0.423 **	−0.193	−0.217 *	0.801 **	1	
Chewing rate	−0.211	−0.287 **	−0.227 *	−0.208	0.015	−0.312 **	−0.390 **	−0.317 **	−0.379 **	0.226 *	1
SW	0.362 **	0.225 *	0.130	0.432 **	−0.369 **	0.271 *	0.493 **	0.727 **	0.465 **	0.351 **	−0.259 *

DMFT, decayed, missing, and filled teeth; SW, swallowing threshold. * *p* < 0.05; ** *p* < 0.01.

**Table 4 nutrients-14-02614-t004:** Cross-tabulation analysis between the development of swallowing threshold and sex and health-related variables.

Participants (*n* = 83)	Normal (%) (*n* = 67)	Developmental Failure (%) (*n* = 16)	*χ* ^2^	*p*-Value
Sex				
Female	36 (53.7)	8 (50.0)		
Male	31 (46.3)	8 (50.0)		
			―	1.000 ^†^
Degree of obesity				
Normal	34 (50.7)	5 (31.3)		
Underweight/severely underweight	27 (40.3)	5 (31.3)		
Overweight/obese	6 (9.0)	6 (37.5)		
			8.598	0.014 ^‡^
Skipping breakfast				
Less than two times a week	48 (71.6)	10 (62.5)		
Two times or more a week	19 (28.4)	6 (37.5)		
			―	0.333 ^†^
Eating between meals				
Less than once a day	31 (46.3)	2 (12.5)		
Once or more a day	36 (53.7)	14 (87.5)		
			―	0.021 ^†^
Physical activity				
30 min or more a day	38 (56.7)	2 (12.5)		
Less than 30 min a day	8 (11.9)	3 (18.8)		
None	21 (31.3)	11 (68.8)		
			10.379	0.006 ^‡^
Self-assessed sleep quality				
Good	66 (98.5)	16 (100)		
Poor	1 (1.5)	0 (0.0)		
			―	1.000 ^†^

^†^ Fisher’s exact test; ^‡^ Chi-square test.

**Table 5 nutrients-14-02614-t005:** Comparison of the development of swallowing threshold on age, height, body weight, dental status, hand grip strength, and masticatory function.

	Normal (*n* = 67)	Developmental Failure (*n* = 16)
Age (years)	9.93 ± 3.09	7.88 ± 3.07 *
Height (m)	1.37 ± 0.18	1.28 ± 0.23
Body weight (kg)	33.41 ± 15.23	29.59 ± 17.10
Number of erupted teeth (N)	23.88 ± 2.96	21.81 ± 2.66 *
DMFT index	4.69 ± 3.91	7.31 ± 4.73 *
Hand grip strength (kg)	17.14 ± 10.39	12.99 ± 9.36
Maximum occlusal force (kN)	0.35 ± 0.10	0.27 ± 0.08 *
Masticatory performance (mg/dL)	134.19 ± 40.75	97.94 ± 24.97 *
Number of chewing cycles (N)	24.64 ± 6.95	15.69 ± 3.93 *
Chewing time (s)	20.91 ± 5.56	14.42 ± 3.97 *
Chewing rate (s/N)	0.86 ± 0.16	0.92 ± 0.13
Swallowing threshold (mg/dL)	134.40 ± 35.61	73.94 ± 10.49 *

Data are expressed as mean ± standard deviation. DMFT, decayed, missing, and filled teeth; Swallowing threshold, glucose concentration on first swallow. Differences between normal and low groups were assessed by a two-tailed *t*-test. * *p* < 0.05 vs. normal group.

**Table 6 nutrients-14-02614-t006:** Predictors of developmental failure of swallowing threshold based on binary logistic regression analysis.

Independent Variables	Category	Adjusted Odds Ratio (95% CI)	*p*-Value
Degree of obesity	Normal	1	―
	Underweight/severely underweight	1.278 (0.323–5.055)	0.727
	Overweight/obese	5.343 (1.168–24.437)	0.031
Eating between meals	Less than once a day	1	―
	Once or more a day	4.934 (1.004–24.244)	0.049

Forward selection (conditional) method. −2 Log likelihood = 69.316. Hosmer and Lemeshow test: *χ*^2^ = 0.719, *p* = 0.949. Cox-Snell R^2^ = 0.135. Nagelkerke R^2^ = 0.216. CI, confidence interval.

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
