# Peer review of "Effects of Developmental Failure of Swallowing Threshold on Obesity and Eating Behaviors in Children Aged 5–15 Years"

_nutrients, 2022, doi:10.3390/nu14132614_

Round 1
Reviewer 1 Report
-What is the hypothesis?
-What is sample size calculus?
-Figure 1-2 may be removed.
-What is food intake of children?
Author Response
The comments have been helpful for us to revise our manuscript. We have therefore thoroughly revised the manuscript on the basis of your suggestions. The parts of the manuscript that have been significantly revised are shown in red font.
- What is the hypothesis?
The hypotheses were added and highlighted in red font in Introduction section.
- What is sample size calculus?
Using software (PS: Power and Sample Size Calculation, available from the Vanderbilt University’s web site), sample size was calculated. I added this sentence in the text.
- Figure 1-2 may be removed.
In accordance with your suggestions, Figures1 and 2 were removed.
- What is food intake of children?
I added the descriptions of the food intake of children in Discussion section.

Reviewer 2 Report
This paper aims to identify factors related to developmental failure of swallowing threshold in children aged 5–15 years. The article is well structure and writing, and provides novel insights into the impacts of nutrition on human health.
Author Response
Response to Reviewer 2 comments
This paper aims to identify factors related to developmental failure of swallowing threshold in children aged 5–15 years. The article is well structure and writing, and provides novel insights into the impacts of nutrition on human health.
Thank you very much for your generous comments on my manuscript.
Reviewer 3 Report
This study has a relevant aim. However, several issues are highlighted.
-Abstract: no number in results were provided.
-Introduction: No hypothesis is provided.
-Methods: This small sample size did not allow factors related to developmental failure of swallowing threshold in children. In addition, evaluating 83 children is very weak to use the logistic regression as statistical analysis.
-Results: Results from tables and figures are very similar and did not convince the importance of manuscript. Additionally, bivariate correlation did not allow to establish the relation of cause and effect between studied variables.
Author Response
Response to Reviewer 3 comments
The comments have been helpful for us to revise our manuscript. We have therefore thoroughly revised the manuscript on the basis of your suggestions. The parts of the manuscript that have been significantly revised are shown in red font.
- Abstract: no number in results were provided.
In accordance with your suggestions, I added the number in results in Abstract section.
- Introduction: No hypothesis is provided.
The hypotheses were add and highlighted in red font in Introduction section.
- Methods: This small sample size did not allow factors related to developmental failure of swallowing threshold in children. In addition, evaluating 83 children is very weak to use the logistic regression as statistical analysis.
As you pointed out, the number of participants was small for performing logistic regression analysis. I added the following description to the paragraph of study limitations.
First, the number of participants was small for performing logistic regression analysis. Among the patients who visited two dental clinics, the number of patients who met all of the conditions we set was very limited. Therefore, this study was performed with the minimum required number of participants calculated by the power analysis. It may be that a highly accurate model will be built in studies with higher numbers of participants.
- Results: Results from tables and figures are very similar and did not convince the importance of manuscript. Additionally, bivariate correlation did not allow to establish the relation of cause and effect between studied variables.
In accordance with your suggestions, Figure 1-2 were removed. I corrected the description of the results in bivariate correlation.

Round 2
Reviewer 2 Report
The small changes made by the authors have improved the manuscript. Thank you and congratulations.
Author Response
Thank you for your generous comments.
Reviewer 3 Report
paper corrected
Author Response
Thank you for your accurate comments.